# Sample-Efficient Reinforcement Learning with Stochastic Ensemble Value Expansion

**Jacob Buckman**[*]   **Danijar Hafner**   **George Tucker**   **Eugene Brevdo**   **Honglak Lee**
Google Brain, Mountain View, CA, USA
`jacobbuckman@gmail.com`, `mail@danijar.com`,
`{gjt,ebrevdo,honglak}@google.com`

## Abstract

Integrating model-free and model-based approaches in reinforcement learning has the potential to achieve the high performance of model-free algorithms with low sample complexity. However, this is difficult because an imperfect dynamics model can degrade the performance of the learning algorithm, and in sufficiently complex environments, the dynamics model will almost always be imperfect. As a result, a key challenge is to combine model-based approaches with model-free learning in such a way that errors in the model do not degrade performance. We propose stochastic ensemble value expansion (STEVE), a novel model-based technique that addresses this issue. By dynamically interpolating between model rollouts of various horizon lengths for each individual example, STEVE ensures that the model is only utilized when doing so does not introduce significant errors. Our approach outperforms model-free baselines on challenging continuous control benchmarks with an order-of-magnitude increase in sample efficiency, and in contrast to previous model-based approaches, performance does not degrade in complex environments.

## 1   Introduction

Deep model-free reinforcement learning has had great successes in recent years, notably in playing video games [23] and strategic board games [27]. However, training agents using these algorithms requires tens to hundreds of millions of samples, which makes many practical applications infeasible, particularly in real-world control problems (e.g., robotics) where data collection is expensive.

Model-based approaches aim to reduce the number of samples required to learn a policy by modeling the dynamics of the environment. A dynamics model can be used to increase sample efficiency in various ways, including training the policy on rollouts from the dynamics model [28], using rollouts to improve targets for temporal difference (TD) learning [7], and using information gained from rollouts as inputs to the policy [31]. Model-based algorithms such as PILCO [4] have shown that it is possible to learn from orders-of-magnitude fewer samples.

These successes have mostly been limited to environments where the dynamics are simple to model. In noisy, complex environments, it is difficult to learn an accurate model of the environment. When the model makes mistakes in this context, it can cause the wrong policy to be learned, hindering performance. Recent work has begun to address this issue. Kalweit and Boedecker [17] train a model-free algorithm on a mix of real and imagined data, adjusting the proportion in favor of real data as the Q-function becomes more confident. Kurutach et al. [20] train a model-free algorithm on purely imaginary data, but use an ensemble of environment models to avoid overfitting to errors made by any individual model.

---

[*]This work was completed as part of the Google AI Residency program.

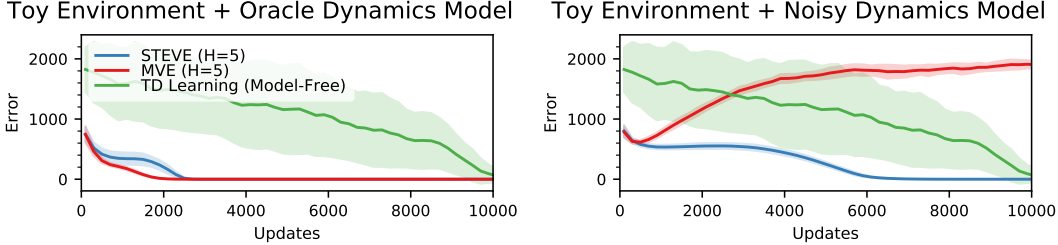

Figure 1: Value error per update on a value-estimation task (fixed policy) in a toy environment. $H$ is the maximum rollout horizon (see Section 3). When given access to a perfect dynamics model, hybrid model-free model-based approaches (MVE and STEVE) solve this task with $5\times$ fewer samples than model-free TD learning. However, when only given access to a noisy dynamics model, MVE diverges due to model errors. In contrast, STEVE converges to the correct solution, and does so with a $2\times$ speedup over TD learning. This is because STEVE dynamically adapts its rollout horizon to accommodate model error. See Appendix A for more details.

We propose *stochastic ensemble value expansion* (STEVE), an extension to *model-based value expansion* (MVE) proposed by Feinberg et al. [7]. Both techniques use a dynamics model to compute "rollouts" that are used to improve the targets for temporal difference learning. MVE rolls out a fixed length into the future, potentially accumulating model errors or increasing value estimation error along the way. In contrast, STEVE interpolates between many different horizon lengths, favoring those whose estimates have lower uncertainty, and thus lower error. To compute the interpolated target, we replace both the model and Q-function with ensembles, approximating the uncertainty of an estimate by computing its variance under samples from the ensemble. Through these uncertainty estimates, STEVE dynamically utilizes the model rollouts only when they do not introduce significant errors. For illustration, Figure 1 compares the sample efficiency of various algorithms on a tabular toy environment, which shows that STEVE significantly outperforms MVE and TD-learning baselines when the dynamics model is noisy. We systematically evaluate STEVE on several challenging continuous control benchmarks and demonstrate that STEVE significantly outperforms model-free baselines with an order-of-magnitude increase in sample efficiency.

## 2   Background

Reinforcement learning aims to learn an agent policy that maximizes the expected (discounted) sum of rewards [29]. The agent starts at an initial state $s_0 \sim p(s_0)$, where $p(s_0)$ is the distribution of initial states of the environment. Then, the agent deterministically chooses an action $a_t$ according to its policy $\pi_\phi(s_t)$ with parameters $\phi$, deterministically transitions to a subsequent state $s_{t+1}$ according to the Markovian dynamics $T(s_t, a_t)$ of the environment, and receives a reward $r_t = r(s_t, a_t, s_{t+1})$. This generates a trajectory of states, actions, and rewards $\tau = (s_0, a_0, r_0, s_1, a_1, \ldots)$. If a trajectory reaches a terminal state, it concludes without further transitions or rewards; however, this is optional, and trajectories may instead be infinite in length. We abbreviate the trajectory by $\tau$. The goal is to maximize the expected discounted sum of rewards along sampled trajectories $J(\theta) = \mathbb{E}_{s_0} \left[ \sum_{t=0}^{\infty} \gamma^t r_t \right]$ where $\gamma \in [0, 1)$ is a discount parameter.

### 2.1   Value Estimation with TD-learning

The action-value function $Q^\pi(s_0, a_0) = \sum_{t=0}^{\infty} \gamma^t r_t$ is a critical quantity to estimate for many learning algorithms. Using the fact that $Q^\pi(s, a)$ satisfies a recursion relation

$$Q^\pi(s, a) = r(s, a) + \gamma(1 - d(s'))Q^\pi(s', \pi(s')),$$

where $s' = T(s, a)$ and $d(s')$ is an indicator function which returns 1 when $s'$ is a terminal state and 0 otherwise. We can estimate $Q^\pi(s, a)$ off-policy with collected transitions of the form $(s, a, r, s')$ sampled uniformly from a replay buffer [29]. We approximate $Q^\pi(s, a)$ with a deep neural network, $\hat{Q}^\pi_\theta(s, a)$. We learn parameters $\theta$ to minimize the mean squared error (MSE) between Q-value

estimates of states and their corresponding TD targets:

$$\mathcal{T}^{TD}(r, s') = r + \gamma(1 - d(s'))\hat{Q}^{\pi}_{\theta^-}(s', \pi(s')) \tag{1}$$

$$\mathcal{L}_{\theta} = \mathbb{E}_{(s,a,r,s')} \left[ (\hat{Q}^{\pi}_{\theta}(s, a) - \mathcal{T}^{\text{TD}}(r, s'))^2 \right] \tag{2}$$

This expectation is taken with respect to transitions sampled from our replay buffer. Note that we use an older copy of the parameters, $\theta^-$, when computing targets [23].

Since we evaluate our method in a continuous action space, it is not possible to compute a policy from our Q-function by simply taking $\max_a \hat{Q}^{\pi}_{\theta}(s, a)$. Instead, we use a neural network to approximate this maximization function [21], by learning a parameterized function $\pi_{\phi}$ to minimize the negative Q-value:

$$\mathcal{L}_{\phi} = -\hat{Q}^{\pi}_{\theta}(s, \pi_{\phi}(s)). \tag{3}$$

In this work, we use DDPG as the base learning algorithm, but our technique is generally applicable to other methods that use TD objectives.

## 2.2 Model-Based Value Expansion (MVE)

Recently, Feinberg et al. [7] showed that a learned dynamics model can be used to improve value estimation. MVE forms TD targets by combining a short term value estimate formed by unrolling the model dynamics and a long term value estimate using the learned $\hat{Q}^{\pi}_{\theta^-}$ function. When the model is accurate, this reduces the bias of the targets, leading to improved performance.

The learned dynamics model consists of three learned functions: the transition function $\hat{T}_{\xi}(s, a)$, which returns a successor state $s'$; a termination function $\hat{d}_{\xi}(s)$, which returns the probability that $s$ is a terminal state; and the reward function $\hat{r}_{\psi}(s, a, s')$, which returns a scalar reward. This model is trained to minimize

$$\mathcal{L}_{\xi,\psi} = \mathbb{E}_{(s,a,r,s')} \left[ ||\hat{T}_{\xi}(s, a) - s'||^2 + \mathbb{H}\left(d(s'), \hat{d}_{\xi}(\hat{T}_{\xi}(s, a))\right) + (\hat{r}_{\psi}(s, a, s') - r)^2 \right], \tag{4}$$

where the expectation is over collected transitions $(s, a, r, s')$, and $\mathbb{H}$ is the cross-entropy. In this work, we consider continuous environments; for discrete environments, the first term can be replaced by a cross-entropy loss term.

To incorporate the model into value estimation, Feinberg et al. [7] replace the standard Q-learning target with an improved target, $\mathcal{T}^{\text{MVE}}_H$, computed by rolling the learned model out for $H$ steps.

$$s'_0 = s', \qquad a'_i = \pi_{\phi}(s'_i), \qquad s'_i = \hat{T}_{\xi}(s'_{i-1}, a'_{i-1}), \qquad D^i = d(s') \prod_{j=1}^{i} (1 - \hat{d}_{\xi}(s'_j)) \tag{5}$$

$$\mathcal{T}^{\text{MVE}}_H(r, s') = r + \left( \sum_{i=1}^{H} D^i \gamma^i \hat{r}_{\psi}(s'_{i-1}, a'_{i-1}, s'_i) \right) + D^{H+1} \gamma^{H+1} \hat{Q}^{\pi}_{\theta^-}(s'_H, a'_H). \tag{6}$$

To use this target, we substitute $\mathcal{T}^{\text{MVE}}_H$ in place of $\mathcal{T}^{\text{TD}}$ when training $\theta$ using Equation 2.[2] Note that when $H = 0$, MVE reduces to TD-learning (i.e., $\mathcal{T}^{\text{TD}} = \mathcal{T}^{\text{MVE}}_0$).

When the model is perfect and the learned Q-function has similar bias on all states and actions, Feinberg et al. [7] show that the MVE target with rollout horizon $H$ will decrease the target error by a factor of $\gamma^{2H}$. Errors in the learned model can lead to worse targets, so in practice, we must tune $H$ to balance between the errors in the model and the $Q$-function estimates. An additional challenge is that the bias in the learned Q-function is not uniform across states and actions [7]. In particular,

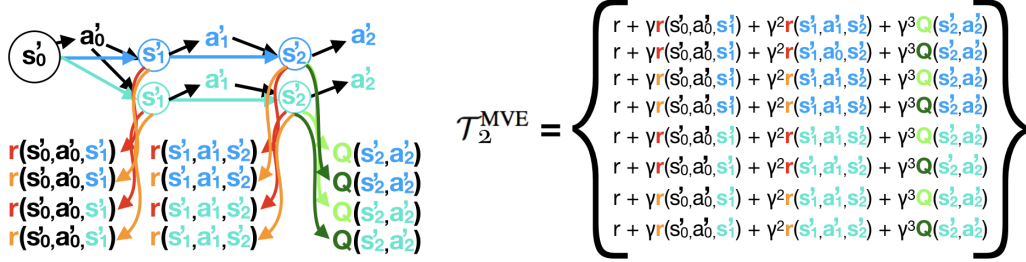

Figure 2: Visualization of how the set of possible values for each candidate target is computed, shown for a length-two rollout with $M, N, L = 2$. Colors correspond to ensemble members. Best viewed in color.

they find that the bias in the Q-function on states sampled from the replay buffer is lower than when the Q-function is evaluated on states generated from model rollouts. They term this the *distribution mismatch problem* and propose the *TD-k trick* as a solution; see Appendix B for further discussion of this trick.

While the results of Feinberg et al. [7] are promising, they rely on task-specific tuning of the rollout horizon $H$. This sensitivity arises from the difficulty of modeling the transition dynamics and the $Q$-function, which are task-specific and may change throughout training as the policy explores different parts of the state space. Complex environments require much smaller rollout horizon $H$, which limits the effectiveness of the approach (e.g., Feinberg et al. [7] used $H = 10$ for HalfCheetah-v1, but had to reduce to $H = 3$ on Walker2d-v1). Motivated by this limitation, we propose an approach that balances model error and Q-function error by dynamically adjusting the rollout horizon.

## 3   Stochastic Ensemble Value Expansion

From a single rollout of $H$ timesteps, we can compute $H+1$ distinct *candidate targets* by considering rollouts of various horizon lengths: $\mathcal{T}_0^{\text{MVE}}, \mathcal{T}_1^{\text{MVE}}, \mathcal{T}_2^{\text{MVE}}, ..., \mathcal{T}_H^{\text{MVE}}$. Standard TD learning uses $\mathcal{T}_0^{\text{MVE}}$ as the target, while MVE uses $\mathcal{T}_H^{\text{MVE}}$ as the target. We propose interpolating all of the candidate targets to produce a target which is better than any individual. Conventionally, one could average the candidate targets, or weight the candidate targets in an exponentially-decaying fashion, similar to TD($\lambda$) [29]. However, we show that we can do still better by weighting the candidate targets in a way that balances errors in the learned $Q$-function and errors from longer model rollouts. STEVE provides a computationally-tractable and theoretically-motivated algorithm for choosing these weights. We describe the algorithm for STEVE in Section 3.1, and justify it in Section 3.2.

### 3.1   Algorithm

To estimate uncertainty in our learned estimators, we maintain an ensemble of parameters for our Q-function, reward function, and model: $\boldsymbol{\theta} = \{\theta_1, ..., \theta_L\}$, $\boldsymbol{\psi} = \{\psi_1, ..., \psi_N\}$, and $\boldsymbol{\xi} = \{\xi_1, ..., \xi_M\}$, respectively. Each parameterization is initialized independently and trained on different subsets of the data in each minibatch.

We roll out an $H$ step trajectory with each of the $M$ models, $\tau^{\xi_1}, ..., \tau^{\xi_M}$. Each trajectory consists of $H + 1$ states, $\tau_0^{\xi_m}, ..., \tau_H^{\xi_m}$, which correspond to $s'_0, ..., s'_H$ in Equation 5 with the transition function parameterized by $\xi_m$. Similarly, we use the $N$ reward functions and $L$ Q-functions to evaluate Equation 6 for each $\tau^{\xi_m}$ at every rollout-length $0 \leq i \leq H$. This gives us $M \cdot N \cdot L$ different values of $\mathcal{T}_i^{\text{MVE}}$ for each rollout-length $i$. See Figure 2 for a visualization of this process.

Using these values, we can compute the empirical mean $\mathcal{T}_i^{\mu}$ and variance $\mathcal{T}_i^{\sigma^2}$ for each partial rollout of length $i$. In order to form a single target, we use an inverse variance weighting of the means:

$$\mathcal{T}_H^{\text{STEVE}}(r, s') = \sum_{i=0}^{H} \frac{\tilde{w}_i}{\sum_j \tilde{w}_j} \mathcal{T}_i^{\mu}, \qquad \tilde{w}_i^{-1} = \mathcal{T}_i^{\sigma^2} \tag{7}$$

To learn a value function with STEVE, we substitute in $\mathcal{T}_H^{\mathrm{STEVE}}$ in place of $\mathcal{T}^{\mathrm{TD}}$ when training $\theta$ using Equation 2.

### 3.2 Derivation

We wish to find weights $w_i$, where $\sum_i w_i = 1$ that minimize the mean-squared error between the weighted-average of candidate targets $\mathcal{T}_0^{\mathrm{MVE}}, \mathcal{T}_1^{\mathrm{MVE}}, \mathcal{T}_2^{\mathrm{MVE}}, ..., \mathcal{T}_H^{\mathrm{MVE}}$ and the true Q-value.

$$\mathbb{E}\left[\left(\sum_{i=0}^{H} w_i \mathcal{T}_i^{\mathrm{MVE}} - Q^\pi(s,a)\right)^2\right] = \mathrm{Bias}\left(\sum_i w_i \mathcal{T}_i^{\mathrm{MVE}}\right)^2 + \mathrm{Var}\left(\sum_i w_i \mathcal{T}_i^{\mathrm{MVE}}\right)$$

$$\approx \mathrm{Bias}\left(\sum_i w_i \mathcal{T}_i^{\mathrm{MVE}}\right)^2 + \sum_i w_i^2 \, \mathrm{Var}(\mathcal{T}_i^{\mathrm{MVE}}),$$

where the expectation considers the candidate targets as random variables conditioned on the collected data and minibatch sampling noise, and the approximation is due to assuming the candidate targets are independent[3].

Our goal is to minimize this with respect to $w_i$. We can estimate the variance terms using empirical variance estimates from the ensemble. Unfortunately, we could not devise a reliable estimator for the bias terms, and this is a limitation of our approach and an area for future work. In this work, we ignore the bias terms and minimize the weighted sum of variances

$$\sum_i w_i^2 \, \mathrm{Var}(\mathcal{T}_i^{\mathrm{MVE}}).$$

With this approximation, which is equivalent to in inverse-variance weighting [8], we achieve state-of-the-art results. Setting each $w_i$ equal to $\frac{1}{\mathrm{Var}(\mathcal{T}_i^{\mathrm{MVE}})}$ and normalizing yields the formula for $\mathcal{T}_H^{\mathrm{STEVE}}$ given in Equation 7.

### 3.3 Note on ensembles

This technique for calculating uncertainty estimates is applicable to any family of models from which we can sample. For example, we could train a Bayesian neural network for each model [22], or use dropout as a Bayesian approximation by resampling the dropout masks each time we wish to sample a new model [10]. These options could potentially give better diversity of various samples from the family, and thus better uncertainty estimates; exploring them further is a promising direction for future work. However, we found that these methods degraded the accuracy of the base models. An ensemble is far easier to train, and so we focus on that in this work. This is a common choice, as the use of ensembles in the context of uncertainty estimations for deep reinforcement learning has seen wide adoption in the literature. It was first proposed by Osband et al. [25] as a technique to improve exploration, and subsequent work showed that this approach gives a good estimate of the uncertainty of both value functions [17] and models [20].

## 4 Experiments

### 4.1 Implementation

We use DDPG [21] as our baseline model-free algorithm. We train two deep feedforward neural networks, a Q-function network $\hat{Q}_\theta^\pi(s,a)$ and a policy network $\pi_\phi(s)$, by minimizing the loss functions given in Equations 2 and 3. We also train another three deep feedforward networks to represent our world model, corresponding to function approximators for the transition $\hat{T}_\xi(s,a)$, termination $\hat{d}_\xi(t \mid s)$, and reward $\hat{r}_\psi(s,a,s')$, and minimize the loss function given in Equation 4.

When collecting rollouts for evaluation, we simply take the action selected by the policy, $\pi_\phi(s)$, at every state $s$. (Note that only the policy is required at test-time, not the ensembles of Q-functions,

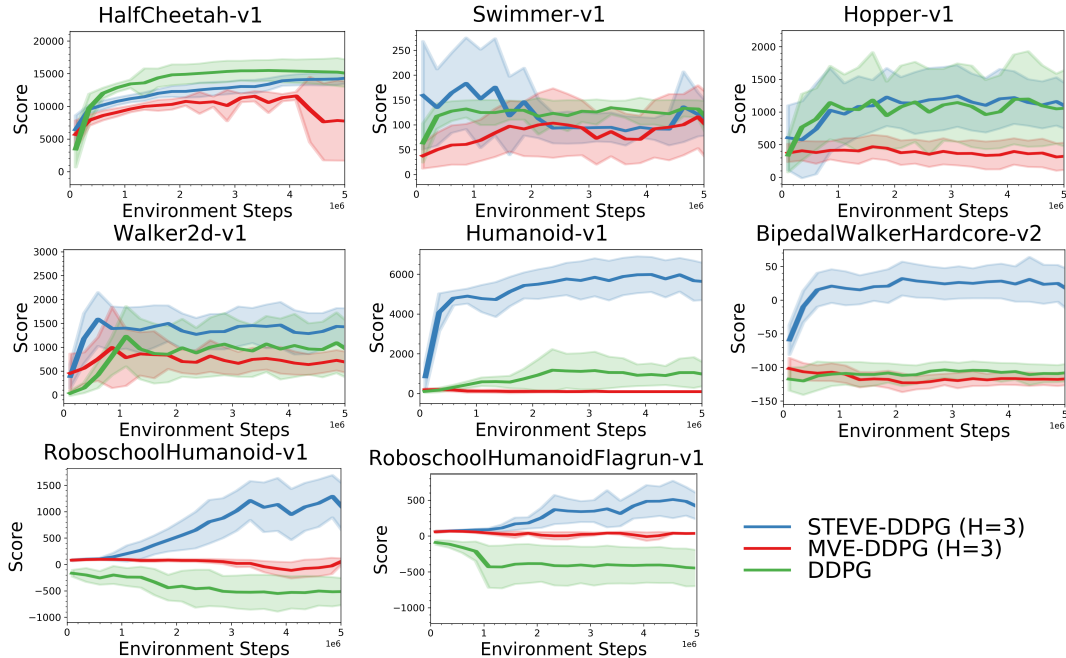

Figure 3: Learning curves comparing sample efficiency of our method to both model-free and model-based baselines. Each experiment was run four times.

dynamics models, or reward models.) Each run was evaluated after every 500 updates by computing the mean total episode reward (referred to as score) across many environment restarts. To produce the lines in Figures 3, 4, and 5, these evaluation results were downsampled by splitting the domain into non-overlapping regions and computing the mean score within each region across several runs. The shaded area shows one standard deviation of scores in the region as defined above.

When collecting rollouts for our replay buffer, we do $\epsilon$-greedy exploration: with probability $\epsilon$, we select a random action by adding Gaussian noise to the pre-tanh policy action.

All algorithms were implemented in Tensorflow [1]. We use a distributed implementation to parallelize computation. In the style of ApeX [16], IMPALA [6], and D4PG [2], we use a centralized learner with several agents operating in parallel. Each agent periodically loads the most recent policy, interacts with the environment, and sends its observations to the central learner. The learner stores received frames in a replay buffer, and continuously loads batches of frames from this buffer to use as training data for a model update. In the algorithms with a model-based component, there are two learners: a policy-learner and a model-learner. In these cases, the policy-learner periodically reloads the latest copy of the model.

All baselines reported in this section were re-implementations of existing methods. This allowed us to ensure that the various methods compared were consistent with one another, and that the differences reported are fully attributable to the independent variables in question. Our baselines are competitive with state-of-the-art implementations of these algorithms [7, 14]. All MVE experiments utilize the TD-k trick. For hyperparameters and additional implementation details, please see Appendix C.[4]

## 4.2 Comparison of Performance

We evaluated STEVE on a variety of continuous control tasks [3, 19]; we plot learning curves in Figure 3. We found that STEVE yields significant improvements in both performance and sample efficiency across a wide range of environments. Importantly, the gains are most substantial in the complex environments. On the most challenging environments: Humanoid-v1, RoboschoolHumanoid-

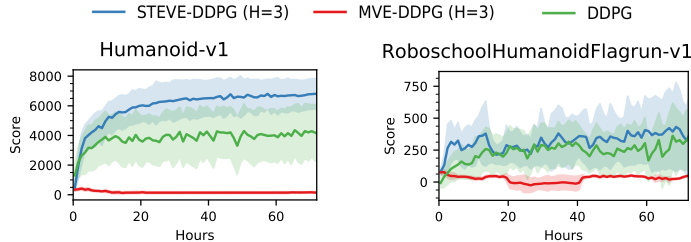

Humanoid-v1          RoboschoolHumanoidFlagrun-v1

Figure 5: Comparison of wall-clock time between our method and baselines. Each experiment was run three times.

v1, RoboschoolHumanoidFlagrun-v1, and BipedalWalkerHardcore-v2, STEVE is the only algorithm to show significant learning within 5M frames.

## 4.3 Ablation Study

In order to verify that STEVE's gains in sample efficiency are due to the reweighting, and not simply due to the additional parameters of the ensembles of its components, we examine several ablations. Ensemble MVE is the regular MVE algorithm, but the model and Q-functions are replaced with with ensembles. Mean-MVE uses the exact same architecture as STEVE, but uses a simple uniform weighting instead of the uncertainty-aware reweighting scheme. Similarly, TDL25 and TDL75 correspond to $TD(\lambda)$ reweighting schemes with $\lambda = 0.25$ and $\lambda = 0.75$, respectively. COV-STEVE is a version of STEVE which includes the covariances between candidate targets when computing the weights (see Section 3.2). We also investigate the effect of the horizon parameter on the performance of both STEVE and MVE. These results are shown in Figure 4.

All of these variants show the same trend: fast initial gains, which quickly taper off and are overtaken by the baseline. STEVE is the only

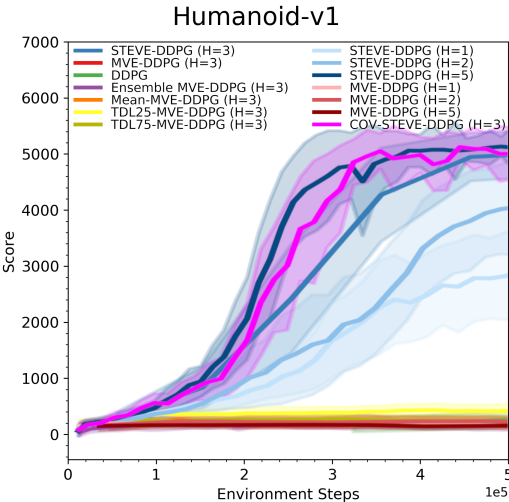

Figure 4: Ablation experiments on variation of methods. Each experiment was run twice.

variant to converge faster and higher than the baseline; this provides strong evidence that the gains come specifically from the uncertainty-aware reweighting of targets. Additionally, we find that increasing the rollout horizon increases the sample efficiency of STEVE, even though the dynamics model for Humanoid-v1 has high error.

## 4.4 Wall-Clock Comparison

In the previous experiments, we synchronized data collection, policy updates, and model updates. However, when we run these steps asynchronously, we can reduce the wall-clock time at the risk of instability. To evaluate this configuration, we compare DDPG, MVE-DDPG, and STEVE-DPPG on Humanoid-v1 and RoboschoolHumanoidFlagrun-v1. Both were trained on a P100 GPU and had 8 CPUs collecting data; STEVE-DPPG additionally used a second P100 to learn a model in parallel. We plot reward as a function of wall-clock time for these tasks in Figure 5. STEVE-DDPG learns more quickly on both tasks, and it achieves a higher reward than DDPG and MVE-DDPG on Humanoid-v1 and performs comparably to DDPG on RoboschoolHumanoidFlagrun-v1. Moreover, in future work, STEVE could be accelerated by parallelizing training of each component of the ensemble.

## 5 Discussion

Our primary experiments (Section 4.2) show that STEVE greatly increases sample efficiency relative to baselines, matching or out-performing both MVE-DDPG and DDPG baselines on every task. STEVE also outperforms other recently-published results on these tasks in terms of sample efficiency [13, 14, 26]. Our ablation studies (Section 4.3) support the hypothesis that the increased performance is due to the uncertainty-dependent reweighting of targets, as well as demonstrate that the performance of STEVE consistently increases with longer horizon lengths, even in complex environments. Finally, our wall-clock experiments (Section 4.4) demonstrate that in spite of the additional computation per epoch, the gains in sample efficiency are enough that it is competitive with model-free algorithms in terms of wall-clock time. The speed gains associated with improved sample efficiency will only be exacerbated as samples become more expensive to collect, making STEVE a promising choice for applications involving real-world interaction.

Given that the improvements stem from the dynamic reweighting between horizon lengths, it may be interesting to examine the choices that the model makes about which candidate targets to favor most heavily. In Figure 6, we plot the average model usage over the course of training. Intriguingly, most of the lines seem to remain stable at around 50% usage, with two notable exceptions: Humanoid-v1, the most complex environment tested (with an observation-space of size 376); and Swimmer-v1, the least complex environment tested (with an observation-space of size 8). This supports the hypothesis that STEVE is trading off between Q-function bias and model bias; it chooses to ignore the model almost immediately when the environment is too complex to learn, and gradually ignores the model as the Q-function improves if an optimal environment model is learned quickly.

## 6 Related Work

Sutton and Barto [29] describe TD($\lambda$), a family of Q-learning variants in which targets from multiple timesteps are merged via exponentially decay. STEVE is similar in that it is also computing a weighted average between targets. However, our approach is significantly more powerful because it adapts the weights to the specific characteristics of each individual rollout, rather than being constant between examples and throughout training. Our approach can be thought of as a generalization of TD($\lambda$), in that the two approaches are equivalent in the specific case where the overall uncertainty grows exponentially at rate $\lambda$ at every timestep.

Munos et al. [24] propose Retrace($\lambda$), a low-variance method for off-policy Q-learning. Retrace($\lambda$) is an off-policy correction method, so it learns from n-step off-policy data by multiplying each term of the loss by a correction coefficient, the *trace*, in order to re-weight the data distribution to look more like the on-policy distribution. Specifically, at each timestep, Retrace($\lambda$) updates the coefficient for that term by multiplying it by $\lambda \mathtt{min}(1, \frac{\pi(a_s|x_s)}{\mu(a_s|x_s)})$. Similarly to TD($\lambda$), the $\lambda$ parameter corresponds to an exponential decay of the weighting of potential targets. STEVE approximates this weighting in a more complex way, and additionally learns a predictive model of the environment (under which on-policy rollouts are possible) instead of using off-policy correction terms to re-weight real off-policy rollouts.

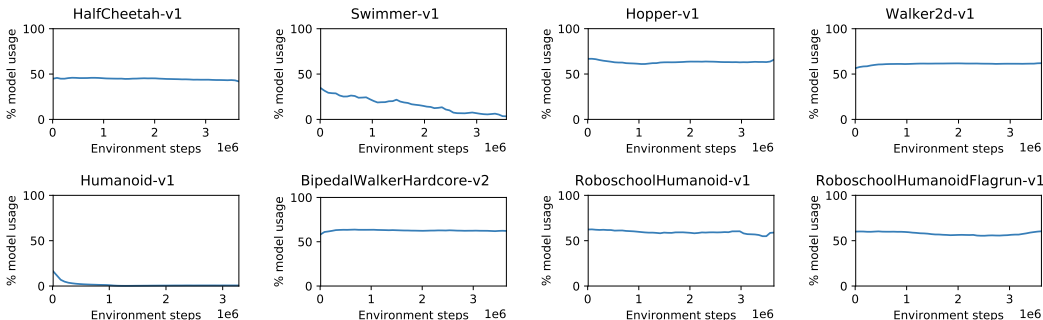

Figure 6: Average model usage for STEVE on each environment. The y-axis represents the amount of probability mass assigned to weights that were not $w_0$, i.e. the probability mass assigned to candidate targets that include at least one step of model rollout.

Heess et al. [15] describe *stochastic value gradient* (SVG) methods, which are a general family of hybrid model-based/model-free control algorithms. By re-parameterizing distributions to separate out the noise, SVG is able to learn stochastic continuous control policies in stochastic environments. STEVE currently operates only with deterministic policies and environments, but this is a promising direction for future work.

Kurutach et al. [20] propose *model-ensemble trust-region policy optimization* (ME-TRPO), which is motivated similarly to this work in that they also propose an algorithm which uses an ensemble of models to mitigate the deleterious effects of model bias. However, the algorithm is quite different. ME-TRPO is a purely model-based policy-gradient approach, and uses the ensemble to avoid overfitting to any one model. In contrast, STEVE interpolates between model-free and model-based estimates, uses a value-estimation approach, and uses the ensemble to explicitly estimate uncertainty.

Kalweit and Boedecker [17] train on a mix of real and imagined rollouts, and adjust the ratio over the course of training by tying it to the variance of the Q-function. Similarly to our work, this variance is computed via an ensemble. However, they do not adapt to the uncertainty of individual estimates, only the overall ratio of real to imagined data. Additionally, they do not take into account model bias, or uncertainty in model predictions.

Weber et al. [31] use rollouts generated by the dynamics model as inputs to the policy function, by "summarizing" the outputs of the rollouts with a deep neural network. This second network allows the algorithm to implicitly calculate uncertainty over various parts of the rollout and use that information when making its decision. However, I2A has only been evaluated on discrete domains. Additionally, the lack of explicit model use likely tempers the sample-efficiency benefits gained relative to more traditional model-based learning.

Gal et al. [11] use a deep neural network in combination with the PILCO algorithm [4] to do sample-efficient reinforcement learning. They demonstrate good performance on the continuous-control task of cartpole swing-up. They model uncertainty in the learned neural dynamics function using dropout as a Bayesian approximation, and provide evidence that maintaining these uncertainty estimates is very important for model-based reinforcement learning.

Depeweg et al. [5] use a Bayesian neural network as the environment model in a policy search setting, learning a policy purely from imagined rollouts. This work also demonstrates that modeling uncertainty is important for model-based reinforcement learning with neural network models, and that uncertainty-aware models can escape many common pitfalls.

Gu et al. [12] propose a continuous variant of Q-learning known as *normalized advantage functions* (NAF), and show that learning using NAF can be accelerated by using a model-based component. They use a variant of Dyna-Q [28], augmenting the experience available to the model-free learner with imaginary on-policy data generated via environment rollouts. They use an iLQG controller and a learned locally-linear model to plan over small, easily-modelled regions of the environment, but find that using more complex neural network models of the environment can yield errors.

Thomas et al. [30] define the $\Omega$-return, an alternative to the $\lambda$-return that accounts for the variance of, and correlations between, predicted returns at multiple timesteps. Similarly to STEVE, the target used is an unbiased linear combination of returns with minimum variance. However, the $\Omega$-return is not directly computable in non-tabular state spaces, and does n-step off-policy learning rather than learn a predictive model of the environment. Drawing a theoretical connection between the STEVE algorithm and the $\Omega$-return is an interesting potential direction for future work.

# 7    Conclusion

In this work, we demonstrated that STEVE, an uncertainty-aware approach for merging model-free and model-based reinforcement learning, outperforms model-free approaches while reducing sample complexity by an order magnitude on several challenging tasks. We believe that this is a strong step towards enabling RL for practical, real-world applications. Since submitting this manuscript for publication, we have further explored the relationship between STEVE and recent work on overestimation bias [9], and found evidence that STEVE may help to reduce this bias. Other future directions include exploring more complex worldmodels for various tasks, as well as comparing various techniques for calculating uncertainty and estimating bias.

**Acknowledgments**

The authors would like to thank the following individuals for their valuable insights and discussion: David Ha, Prajit Ramachandran, Tuomas Haarnoja, Dustin Tran, Matt Johnson, Matt Hoffman, Ishaan Gulrajani, and Sergey Levine. Also, we would like to thank Jascha Sohl-Dickstein, Joseph Antognini, Shane Gu, and Samy Bengio for their feedback during the writing process, and Erwin Coumans for his help on PyBullet enivronments. Finally, we would like to thank our anonymous reviewers for their insightful suggestions.

## Footnotes

[2]This formulation is a minor generalization of the original MVE objective in that we additionally model the reward function and termination function; Feinberg et al. [7] consider "fully observable" environments in which the reward function and termination condition were known, deterministic functions of the observations. Because we use a function approximator for the termination condition, we compute the accumulated probability of termination, $D^i$, at every timestep, and use this value to discount future returns.

[3]Initial experiments suggested that omitting the covariance cross terms provided significant computational speedups at the cost of a slight performance degradation. As a result, we omitted the terms in the rest of the experiments.

[4]Our code is available open-source at: `https://github.com/tensorflow/models/tree/master/research/steve`

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
