[Supplementary Material]

## A  Toy Problem: A Tabular FSM with Model Noise

To demonstrate the benefits of Bayesian model-based value expansion, we evaluated it on a toy problem. We used a finite state environment with states $\{s_0, \ldots, s_{100}\}$, and a single action $A$ available at every state which always moves from state $s_t$ to $s_{t+1}$, starting at $s_0$ and terminating at $s_{100}$. The reward for every action is -1, except when moving from $s_{99}$ to $s_{100}$, which is +100. Since this environment is so simple, there is only one possible policy $\pi$, which is deterministic. It is possible to compute the true action-value function in closed form, which is $Q^\pi(s_i, A) = i$.

We estimate the value of each state using tabular TD-learning. We maintain a tabular function $\hat{Q}^\pi(s_i, A)$, which is just a lookup table matching each state to its estimated value. We initialize all values to random integers between 0 and 99, except for the terminal state $s_{100}$, which we initialize to 0 (and keep fixed at 0 at all times). We update using the standard undiscounted one-step TD update, $\hat{Q}^\pi(s_i, A) = r + \hat{Q}^\pi(s_{i+1}, A)$. For each update, we sampled a nonterminal state and its corresponding transition $(s_i, r, s_{i+1})$ at random. For experiments with an ensemble of Q-functions, we repeat this update once for each member of the ensemble at each timestep.

The transition and reward function for the oracle dynamics model behaved exactly the same as the true environment. In the "noisy" dynamics model, noise was added in the following way: 10% of the time, rather than correctly moving from $s_t$ to $s_{t+1}$, the model transitions to a random state. (Other techniques for adding noise gave qualitatively similar results.)

On the y-axis of Figure 1, we plot the mean squared error between the predicted values and the true values of each state: $\frac{1}{100} \sum_{i=0}^{99} (\hat{Q}^\pi(s_i, A) - Q^\pi(s_i, A))^2$.

For both the STEVE and MVE experiments, we use an ensemble of size 8 for both the model and the Q-function. To compute the MVE target, we average across all ensembled rollouts and predictions.

## B  The TD-k Trick

The TD-k trick, proposed by Feinberg et al. [7], involves training the Q-function using every intermediate state of the rollout:

$$s'_{-1} = s$$
$$\mathcal{L}_\theta = \mathbb{E}_{(s,a,r,s')} \left[ \frac{1}{H} \sum_{i=-1}^{H-1} (\hat{Q}^\pi_\theta(s'_i, a_i) - \mathcal{T}^{\text{MVE}}_H(r_i, s'_{i+1}))^2 \right],$$

where $s'_i, r_i, a_i$ are defined as in Equation 5.

To summarize Feinberg et al. [7], the TD-k trick is helpful because the off-policy states collected by the replay buffer may have little overlap with the states encountered during on-policy model rollouts. Without the TD-k trick, the Q-function approximator is trained to minimize error only on states collected from the replay buffer, so it is likely to have high error on states not present in the replay buffer. This would imply that the Q-function has high error on states produced by model rollouts, and that this error may in fact continue to increase the more steps of on-policy rollout we take. By invoking the TD-k trick, and training the Q-function on intermediate steps of the rollout, Feinberg et al. [7] show that we can decrease the Q-function bias on frames encountered during model-based rollouts, leading to better targets and improved performance.

The TD-k trick is orthogonal to STEVE. STEVE tends to ignore estimates produced by states with poorly-learned Q-values, so it is not hurt nearly as much as MVE by the distribution mismatch problem. However, better Q-values will certainly provide more information with which to compute STEVE's target, so in that regard the TD-k trick seems beneficial. An obvious question is whether these two approaches are complimentary. STEVE+TD-k is beyond the scope of this work, and we did not give it a rigorous treatment; however, initial experiments were not promising. In future work, we hope to explore the connection between these two approaches more deeply.

## C   Implementation Details

All models were feedforward neural networks with ReLU nonlinearities. The policy network, reward model, and termination model each had 4 layers of size 128, while the transition model had 8 layers of size 512. All environments were reset after 1000 timesteps. Parameters were trained with the Adam optimizer Kingma and Ba [18] with a learning rate of 3e-4.

Policies were trained using minibatches of size 512 sampled uniformly at random from a replay buffer of size 1e6. The first 1e5 frames were sampled via random interaction with the environment; after that, 4 policy updates were performed for every frame sampled from the environment. (In Section 4.4, the policy updates and frames were instead de-synced.) Policy checkpoints were saved every 500 updates; these checkpoints were also frozen and used as $\theta^-$. For model-based algorithms, the most recent checkpoint of the model was loaded every 500 updates as well.

Each policy training had 8 agents interacting with the environment to send frames back to the replay buffer. These agents typically took the greedy action predicted by the policy, but with probability $\epsilon = 0.05$, instead took an action sampled from a normal distribution surrounding the pre-tanh logit predicted by the policy. In addition, each policy had two greedy agents interacting with the environment for evaluation.

Dynamics models were trained using minibatches of size 1024 sampled uniformly at random from a replay buffer of size 1e6. The first 1e5 frames were sampled via random interaction with the environment; the dynamics model was then pre-trained for 1e5 updates. After that, 4 model updates were performed for every frame sampled from the environment. (In Section 4.4, the model updates and frames were instead de-synced.) Model checkpoints were saved every 500 updates.

All ensembles were of size 4. During training, each ensemble member was trained on an independently-sampled minibatch; all minibatches were drawn from the same buffer. Additionally, $M, N, L = 4$ for all experiments.