[Reviews · NeurIPS 2018]

Reviewer 1



# Paper ID 5026 Sample-efficient RL with stochastic ensemble value expansion ## Summary The paper proposes a method to learn a memoryless policy in a sample efficient manner using an ensemble of learned MDP models, policies and Q functions. The main algorithmic idea is a weighted combination of H step temporal differences, estimated on H steps (and rolled out by a learned model of the environment). The underlying idea is to allow the learner to tradeoff between estimation errors in model and Q function in different parts of the state-action space during learning. The updated TD estimator is incorporated into the DDPG algorithm in a straightforward manner. The update is computationally more intensive but the result is improved sample complexity. The experimental results on a variety of continuous control tasks show significant improvement over the baseline DDPG and a related method (MVE) (which is the precursor to this work). Overall, the paper is well written. The empirical results are very promising. The analysis and discussion is a bit limited but is not a major drawback. Overall, there is much to like about the paper. Detailed comments follow. ## Detailed Comments - The main idea is to replace the vanilla temporal difference update with one that uses simulated trajectories to look H steps ahead (followed by the regular TD update on step H+1). The simulated trajectories are generated from an ensemble of M models. Additional ensembles of reward functions and Q functions are maintained during learning. - My understanding is that this additional overhead (M models, N reward functions, L Q-functions) is only required during training. At test time, only the final learned Q function (and policy network) is required (right?). The paper could perhaps better describe the difference between training and test. - In Section 3.1, I'm not sure I completely followed how the mean and variance of the TD estimates at each time step $i$ are computed. I believe the algorithm creates MNL samples on each (s, a) pair visited during training and uses the MNL samples to fit the T_i mean and variance *for that particular (s, a) pair (or (r, s') pair)*. Is this correct? I think Section 3.1 needs to be described much more clearly. A graphical visualization of the N models being rolled out and converted into the final T_i mean and variance estimates may make the learning algorithm easier to visualize. - Assuming I've understood correctly, the proposed algorithm (STEVE) boils down to using the learned models to generate TD estimates at every step $0 \leq i \leq H$. The TD estimates are noisy (due to model errors) so the final TD estimate is a weighted combination of estimated means of the $T_i$, where the weights are larger when the variance in $T_i$ is small. - Overall, the approach makes good intuitive sense to me. The authors provide some theoretical justification of the algorithm. A more rigorous analysis in Section 3.2 would strengthen the paper. - The modified TD update is plugged into the DDPG RL algorithm which uses a pair of neural networks for the Q function and the policy. The paper evaluates the algorithm on a number of continuous control tasks. - The experimental section shows that the results are quite promising. The results show the STEVE algorithm doing significantly better. The baselines are the vanilla DDPG and the MVE variant of DDPG. - In Figure 2, I'd be interested in a discussion of the high variance. Perhaps the caption could include more details about the averaging that was performed? Additional baselines (besides DDPG) would also be good to see on the same chart. - The ablation study description in Section 4.3 was a bit confusing. Why would the "additional parameters of the ensembles" (line 181) matter to the test-time score in Figure 3? This seems to imply that the test time score depends on the ensemble somehow but I don't think that's happening here. What am I missing? - I found Figure 3 to be very interesting. The performance of STEVE across (H=1/3/5) was particularly interesting as well as the uniform weighting scheme. These results seem somewhat central to the claims of the method. GCP budget permitting, I think the the experimental section would be much improved if these graphs were generated for the other control tasks as well. Any preliminary results on these would be interesting to hear about as well. - Figure 5 suggests something interesting is happening but I'm not sure I fully understand the discussion in the paper. Why exactly does the model usage go to zero in the least complex AND the most complex environments? I think it might be useful to analyze and describe these three scenarios (going to zero in easy env, going to zero in complex env, hovering around 50%) in more detail. Finally, how are the weights typically distributed across non-zero timesteps? - Overall, I liked the paper. Additional experiments and baselines, experimental analysis and a clearer description of the algorithm would significantly strengthen the paper. Minor points: - I think the ensemble size ought to be mentioned somewhere in the paper and not the appendix. UPDATE ---------- After reading the other reviews and the author responses, I'm upgrading my evaluation. I congratulate the authors on their fine work.

Reviewer 2



This paper presents an algorithm for reinforcement learning through an improved fusion of model-free and model-based RL. It builds on the model-based value expansion (MVE) algorithm by introducing dynamically-weighted rollout horizons and learned reward and termination functions. The algorithm improves significantly on DDPG in complex simulated environments with better sample efficiency. The work is compelling, showing marked useful improvement over DDPG and MVE. The ablation study is good, although it would be even better to test more than one environment. It is somewhat concerning that the theoretical justification suggests both bias and variance should be taken into account, but bias is ignored because a good estimate was not found. The results with only variance are good enough to justify the omission, but it would be more reassuring if some ideas were given of the magnitude of the bias. The paper is well-organized. It builds up to STEVE in a logical sequence and explains STEVE itself quite clearly. A few places for improvements: In line 22, the abbreviation 'TD' is used without being defined as temporal difference It is not explained until much later what 'H' means in the legend of figure 1. In line 50, what is 'p' in 'p(s_0)'? Before or after equations 3 and 4, it would be helpful to briefly explain in words the function of D. In figure 2, different algorithms have different numbers of steps for the same environment (the green DDPG line is longer than the red MVE line is longer than the blue STEVE line). It would be good to have them all the same length or explain why they're not. Please avoid using red and green in plots as they are difficult for colorblind people to distinguish. Related work is thoroughly cited, and STEVE is unique in adjusting each estimate's weight based on uncertainty. STEVE shows significantly improved adaptability to different environments, as well as good sample efficiency and improved learning rate for some environments. It appears to be a promising algorithm, although the amount of compute necessary may limit its audience.

Reviewer 3



The authors propose an extension of the Model-based Value Expansion (MVE) method for model-augmented reinforcement learning. They observe that a fixed prediction horizon H cause devolving performance in MVE. The interpretation is that errors due to uncertain predictions accumulate in long simulated trajectories, which restricts the algorithm to small H in complex environments. The authors propose Stochastic Ensemble Value Expansion (STEVE), which estimates the uncertainty of model and Q-function with an ensemble method. Instead of a fixed horizon, STEVE computes the targets as a sum of all horizon-targets, weighted by the normalized inverse variance of the ensemble estimate. This automatically adjusts the horizon for each trajectory and implicitly discards uncertain predictions. The algorithm is tested on a variety of continuous control tasks and ablation studies are provided for weighting and wall-clock time. The presented performance is significantly better than the model-free variant DDPG and the MVE variation thereof. The article is well written and the presented extension appears to significantly improve upon MVE. The authors discuss and answer many potential criticisms, for example, STEVE requires a large number of forward-passes through the Q-function but Figure 4 shows that the performance gain is significant in wall-clock time as well. While the amount of experiments is commendable, they also show some inconsistencies that could be better discussed, for example, in the majority of the tasks, the weighting in STEVE remained fairly "stable around 50%" (l.218 and Figure 5). This seems to contradict the bad performance of the ablation "Mean-MVE-DDPG" (and the TD-lambda variants) in Figure 3, which weights the horizon-targets equally. Another example is that in Humanoid-v1 of Figure 5, STEVE converges quickly to DDPG, but the performances of the two algorithms shown in Figures 2 and 3 differ significantly. The reviewer is aware that the contradicting results are not directly comparable, but this should be discussed in more depth. The definition of the TD-k trick in the appendix also appears to be wrong, which would be an alternative explanation for some of the discrepancies. The reviewer recommends to accept the paper with little reservation. The authors are encouraged to check the TD-k trick (and their implementation thereof) and to add some additional discussion of the points raised above. COMMENTS: eq.3: from your description, $D^i$ should depend on $\hat d_\xi(s'_j)$, not $d(s'_j)$. l.80f: $T^MVE_0$ in eq.4 is not equivalent to eq.1, the difference is $D^{H+1}$. I recommend to introduce $d(s')$ earlier and update eq.1 accordingly l.116: clarify that (or if) for each value of $T^MVE_i$ the same reward function is used to compute the rewards of all time steps $j$ ($\hat r_\psi(s'_{j-1}, a'_{j-1}, s'_j$). l.132ff: the optimization problem for the weights $w_i$ can be much more elegantly solved with the Lagrange method (in literally three lines; ignore the implicit constraint $w_i \geq 0$). Interestingly, solving the original approximated problem (including the bias term) yields the same weights plus an additional term that is proportional to the true $Q^\pi(s,a)$. Approximating the true Q-value with the mean estimate could yield the improved weighting you are discussing in l.133. fig.2: for the final version you should run all algorithms for the same number of environment steps. l.170f: did you test the MVE ablations without the TD-k trick (see below why)? appendix.B: the TD-k trick seems wrong. The reviewer is not familiar with the original definition, but in the presented equation the Q-values of all state-action pairs along the imagined trajectory are updated with the *same* target. Intuitively, the target should be $T^MVE_{H-i+1}(r_i, s'_{i+1})$ with appropriately defined $r_i$. This could be a simple typo, but if the implementation is wrong this could also explain why MVE (which you report uses TD-k) performs often so badly in comparison to STEVE (which does not).